# Development of a Transmission Line Model for the Thickness Prediction of Thin Films via the Infrared Interference Method [†]

**Christos Mpilitos [1], Stamatios Amanatiadis [2],\*[iD], Georgios Apostolidis [2][iD],
Theodoros Zygiridis [3], Nikolaos Kantartzis [1][iD] and Georgios Karagiannis [2]**

[1]   Department of Electrical & Computer Engineering, Aristotle University of Thessaloniki,
     GR-54124 Thessaloniki, Greece; mpilitos@ece.auth.gr (C.M.); kant@auth.gr (N.K.)
[2]   Art Diagnosis Center, Ormylia Foundation, GR-63071 Ormylia, Greece; gkaposto@auth.gr (G.A.);
     g.karagiannis@teemail.gr (G.K.)
[3]   Department of Informatics & Telecommunication Engineering, University of Western Macedonia,
     GR-50131 Kozani, Greece; tzygiridis@uowm.gr
\*   Correspondence: samanati@auth.gr; Tel.: +30-6974-228606
[†]   This paper is an extended version of our paper published in Proceedings of the 7th International Conference
     on Modern Circuits and Systems Technologies (MOCAST2018) on Electronics and Communications,
     Thessaloniki, Greece, 7–9 May 2018.

**Abstract:** An efficient transmission line model in the micrometric order is presented in this paper, to determine the thickness of thin dielectric films deposited on highly-doped substrates. In particular, the estimation of the thickness is based on multiple reflections of an incident infrared electromagnetic wave generating interference on the sensor. To this objective, the periodicity of the local maxima and minima, including the phase shift and wavelength dependence of the reflection at the layer-substrate interface, leads in the extraction of the required thickness. Moreover, a theoretical transmission line circuit is designed, in order to model the multiple interferences scenario, and an iterative method is developed to converge towards the correct coating thickness. The featured theoretical transmission line model is validated, via a direct comparison with Certified Reference Materials, to indicate its overall accuracy and reliability level. Finally, the proposed method is utilized to calculate the thickness of coated metallic samples.

**Keywords:** transmission line; near-infrared; interference method; thickness measurement

---

In this work, the infrared interference method and transmission line modeling are combined to evaluate the thickness of thin dielectric films placed atop highly-doped substrates of similar conductivity type. The estimation of this thickness has been deemed a very significant procedure for industrial, biomedical, and even cultural heritage applications [1–6]. Specifically, thin epitaxial films can be located onto several critical objects, in order to prevent their ageing or degradation. As a consequence, accurate and trustworthy calculation of their thickness can indicate their protection efficiency together with other interesting properties, and it has to be performed via non-destructive techniques (NDT) for in situ measurements [7–11].

Several techniques have been hitherto developed for the determination of thin-layer dimensions in the micrometric order, such as acoustic microscopy and optical coherence tomography (OCT) [12–14]. Both of them acquire a tomographic image through the measurement of an incident echo duration (total time of flight); yet, they are prone to various limitations. Explicitly, acoustic microscopy is not able to detect layers thinner than 50 μm, even when utilizing high frequency ultrasounds (approximately at

the frequency of 200 MHz). On the other hand, the OCT method requires an optically transparent film in order to accurately calculate thickness, a fact which increases its complexity and total cost.

The prior limitations may be effectively bypassed through the infrared interference method (IRIM) [15–17], which is based on the interferences that appear in the infrared-spectrum measurement of the considered film, due to the multiple reflections on both of its sides. Ordinarily, the near-infrared regime is observed, because of its nature (which is free of redundant information). Then, the periodicity of the local maxima and minima due to these interferences—taking into account the phase shift and wavelength, in relation to reflection at the interface—can enable the desired thickness prediction.

Most of the existing IRIM models evaluate the thickness through the fundamental reflection theory of electromagnetic waves, considering only the two initial reflections, though. In this context, the current work introduces a transmission line model, equivalent to the original thin-film scenario, for the precise analysis of the entire physical phenomenon. The key novelty and asset of this technique is the incorporation of multiple reflections in the prediction process, thus offering extra degrees of freedom, flexibility, and enhanced precision. Then, an iterative procedure is realized to estimate the correct thin-film thickness, through the minimization of error between the theoretical estimation and the measurement. For its validation, the proposed model is elaborately compared with measurements of Certified Reference Materials (CRM). Finally, some real-life coated metallic samples are measured, and the thickness of their coatings is extracted via the proposed scheme.

## 1. Materials and Methods

### 1.1. Main Concept and Theoretical Formulation

Let us consider the configuration of Figure 1, which depicts the primary geometry of IRIM, and involves two different layered media with refractive indices $n_1, n_2$ and absorption coefficients $a_1, a_2$, respectively. In this context, and neglecting any polarization effects (which are indeed trivial, if the incidence angle $\theta \leq 30°$), the reflection coefficient—taking into account points A and B in Figure 1—is given by

$$R = \frac{r_1^2 + r_2^2 - 2r_1 r_2 \cos(\varphi - \theta)}{1 + r_1^2 r_2^2 - 2r_1 r_2 \cos(\varphi - \theta)}, \tag{1}$$

with $r_1^2$ the reflectivity at the air and first medium interface, $r_2^2$ the reflectivity at the first and second medium interface, and $\varphi$ the phase shift, expressed as

$$r_1^2 = \left(\frac{n_1 - n_0}{n_1 + n_0}\right)^2, \tag{2}$$

$$r_2^2 = \frac{(n_2 - n_1)^2 + a_2^2}{(n_2 + n_1)^2 + a_2^2}, \tag{3}$$

for

$$\varphi = \frac{4\pi n_1 d \sin^2 \theta}{n_1 + n_0} \left(1 - \frac{n_0^2}{n_1^2}\right), \tag{4}$$

$$\theta = \tan^{-1}\left(\frac{2n_1 a_2}{n_1^2 - n_2^2 - a_2^2}\right), \tag{5}$$

$d$ the depth of the first medium, and $n_0$ the air refractive index. Note that the coefficients $n_i$ and $a_i$ (for $i = 1, 2$) are related to the real imaginary part of the complex refraction index, in the sense that $n_c = n - ja$. Furthermore, as already explained, the contribution of polarization effects can be safely omitted, due to the fact that at $\theta = 30°$, the error of reflectivity $r_1^2$ is found to be less than 1% at a wavelength of $\lambda = 50\,\mu\text{m}$. Equivalently, owing to the high value of $n_1$, the incidence angle at the interface between the first and second medium has a maximum of 8.5° at $\theta = 30°$, thus leading to a

less than 0.5% error for reflectivity $r_2^2$. Note that, in the majority of the cases, the sample is illuminated vertically and thus the error is certainly negligible.

Finally, it should be emphasized that the accuracy of the aforementioned technique has proven to be the best, among those commonly employed for semiconductor epitaxially-based material measurements. Moreover, the proposed approach is mainly independent of the shape of the impurity profile. Therefore, it is not necessary to opt for extra profile corrections or rather simplistic conventions for any media variations encountered during the measurement process. The latter deduction naturally implies that much thinner layers can be precisely and reliably evaluated, even when the influence of their profile becomes considerable.

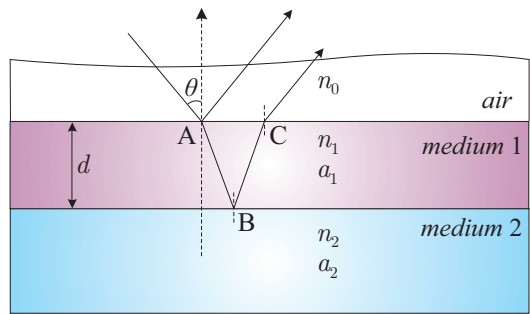

**Figure 1.** Geometric setup and basic principles of the IRIM.

Nevertheless, this simple theoretical analysis accounts only the first of the inner reflections. Although the latter is the most considerable one, the influence of the forthcoming ones can slightly alter the calculations, leading to inaccurate results. To this aim, a new scheme based on transmission line theory has been developed, in order to model the entire electromagnetic phenomenon.

### 1.2. Design and Analysis of the Proposed Model

In our analysis, a thin dielectric film of thickness $d$ is considered, atop a conductor of conductivity $\sigma_c$, as depicted in Figure 2a, since such a scenario is the most commonly analyzed one. The sample is illuminated by an infrared source, and the detector is located at approximately the same spot as the source. Then, the interference of the reflected waves on the upper and lower side of the film, as well as the multiple reflections that occur, appear in the frequency spectrum. However, the type of interference (destructive or constructive) depends on the frequency $\omega$, due to the varying electrical length of the slab thickness. Consequently, several local maxima and minima are introduced in the infrared spectrum, originating from these interferences.

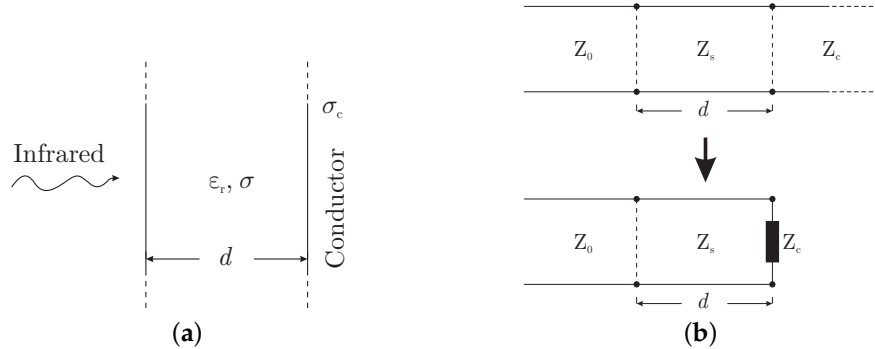

**Figure 2.** (**a**) The configuration under study, consisting of a thin dielectric slab along with the terminating conductor, and (**b**) the featured equivalent transmission line model, for the estimation of the unknown thickness.

In this framework, the periodicity of the extrema is connected to the thickness by the proposed transmission line model, as illustrated in Figure 2b, and is fully equivalent to the real problem. The left part of the transmission line corresponds to air (where $Z_0 = 376.73\,\Omega$), while the central part models the thin film. The rightmost part represents the conductor, considered as an infinite line, as its realistic thickness is significantly larger than the penetration depth, even for poorer conductors. Furthermore, the characteristic impedance $Z_s$ of this region is computed by means of the film characteristics, namely the electric permittivity $\varepsilon$, magnetic permeability $\mu$, and thermal losses $\sigma$, as

$$Z_s = \sqrt{\frac{j\omega\mu}{\sigma + j\omega\varepsilon}}, \tag{6}$$

whereas the complex propagation constant $\gamma$ is

$$\gamma^2 = j\omega\mu(\sigma + j\omega\varepsilon). \tag{7}$$

The characteristic impedance of the conductor, $Z_c$, is calculated equivalently to (6), and is considered safely as the load impedance of the central line, due to the zero reflected wave. Consequently, the analyzed transmission line model is converted to the simpler one of Figure 2b (the lower one). Finally, the input impedance, $Z_{in}$, at the upper layer of the slab, is calculated normally from the transmission line theory as

$$Z_{in} = Z_s \frac{Z_c + Z_s \tanh(\gamma d)}{Z_s + Z_c \tanh(\gamma d)}, \tag{8}$$

while, finally, the reflection coefficient is computed as

$$\Gamma = \frac{Z_{in} - Z_0}{Z_{in} + Z_0}. \tag{9}$$

The matching of the local extrema location of the sample measurement to those of the proposed model leads to the evaluation of the thickness $d$. The complete procedure for this matching is described in the flow chart of Figure 3. Initially, the basic parameters, namely the effective spectral region and a rough estimation of the thickness, are defined. Subsequently, the reflectance is calculated via the proposed transmission line model, and it is compared to the real spectrum of the measurement. A decision is received (based on the reflectance error), and a new estimation of the thickness is realized, and this procedure is repeated until the error is minimized. To this point, the converged thickness value corresponds to the real thickness of the thin film. Note that, at each iteration, the thickness is estimated via optimization algorithms (such as genetic ones).

Additionally, it is important to mention that the multiple reflections are not neglected (as in existing implementations), due to the fundamental properties of transmission line theory, and hence the accuracy of the IRIM method is significantly enhanced. This is actually a key advantage of the featured technique, which can now handle more demanding problems without a need for adopting simplified and rather coarse conventions. Finally, the analysis is more likely to be performed in the near-infrared regime (which is free of redundant information, compared to lower infrared frequencies), where the IR fingerprint of the materials can confuse the algorithm. Nevertheless, there are several cases where the conductor's behaviour is degraded at near-infrared spectrum, especially for impure metals, and the relocation to lower frequencies is unavoidable due to less noisy signals, as depicted in Figure 4.

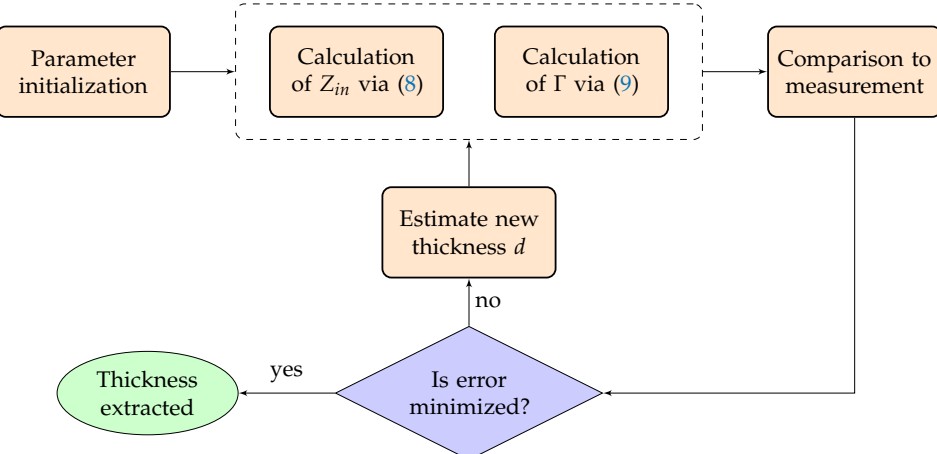

**Figure 3.** Flow chart describing the extraction of the thin film thickness, utilizing the proposed transmission line scheme.

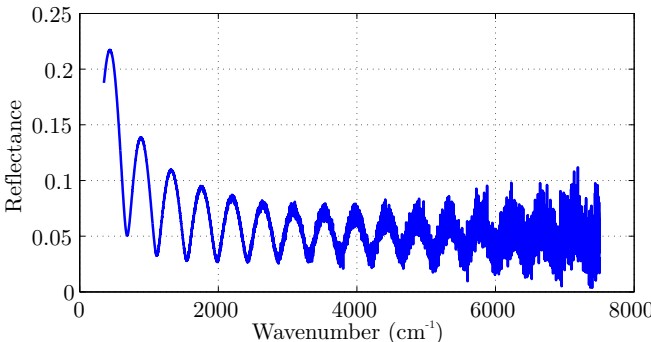

**Figure 4.** An illustration of the periodicity pattern's degradation, due to thermal noise at larger frequencies.

## 2. Results and Discussion

The effectiveness of our method is validated by comparison to the measured spectra of the CRM, depicted in Figure 5. In particular, the genetic algorithm toolbox of MATLAB is utilized in the transmission line model, to extract the thickness that minimizes the comparison error. The total population size of the genetic algorithm is selected to be 200, and one parameter is optimized (namely, the thickness $d$), as the rest of them are considered known. The only enforced restriction is the variable's positive value, while a uniform initial distribution is utilized. The crossover-to-mutation ratio is set to 0.8, while intermediate and uniform of 0.02 rate functions are used, respectively. The samples consist of a ferrous base and a thin-film coating of $\varepsilon = 2.2\varepsilon_0$, $\mu = \mu_0$, and $\sigma = 10^2$ S/m, where $\varepsilon_0$ and $\mu_0$ are the free-space permittivity and permeability, respectively. The thickness of the samples are $d = 15.55$ μm, 39.71 μm, and 98.91 μm, while the Alpha Bruker FTIR is used for the infrared spectra measurements. A total of 32 scans (measurement time of 1 sec each) at 10 different points were realized, by adjusting the sample on the FTIR probe. Although the method can be applied adequately to any sample surface, the utilized ones are of a smooth surface (Figure 5).

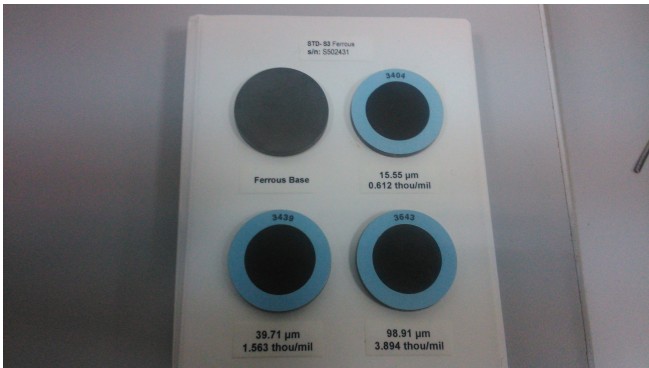

**Figure 5.** The Certified Reference Materials utilized for the validation of the proposed theoretically-designed transmission line model.

In this framework, Figure 6 illustrates the spectrum of the ferrous base, indicating that, at the near infrared regime, the conductor nature degraded. The utilized theoretical conductivity of this material was calculated to match the measurement and its mean value was $\sigma_c = 3221\,\text{S/m}$ with $284\,\text{S/m}$ standard deviation. Due to degradation in the near-infrared spectrum, the region up to $1500\,\text{cm}^{-1}$ is taken into account, where the infrared fingerprint of the material is located; therefore, our estimation may be disturbed, as observed in Figure 7. However, the extrema due to the interferences can be located by thorough examination and considering those that exhibit a periodic pattern. In this way, the accuracy of the method has proven remarkable, since the relative error of the thickness estimation was calculated less than 1% for any sample.

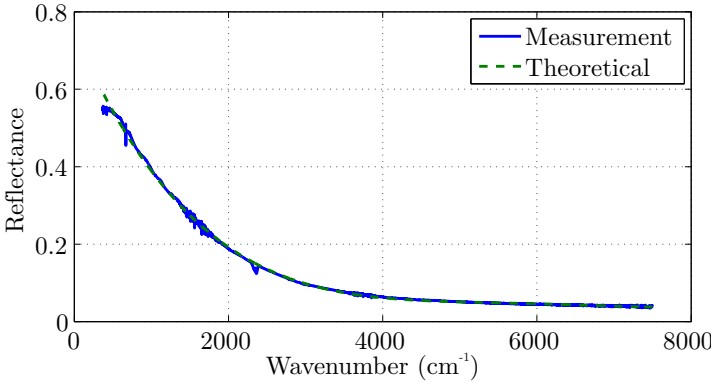

**Figure 6.** The measured versus the theoretical infrared spectrum of the ferrous base underneath the thin dielectric films.

Based on the above observations, the comparison analysis for the thinner sample (namely the one with $d = 15.55\,\mu\text{m}$) is given in Figure 7a, where, although the IR fingerprint peaks significantly degrade the spectrum, the interference peaks can be easily located when considering both maxima and minima. As a consequence, the estimated value is $15.5 \pm 0.32\,\mu\text{m}$. Accordingly, the estimated value for the second sample (i.e., for $d = 39.71\,\mu\text{m}$) is approximately $39.5 \pm 0.77\,\mu\text{m}$, while it is observed from Figure 7b that the interference peaks may be more easily located. Finally, the spectrum of the thicker sample ($d = 98.91\,\mu\text{m}$), is presented in Figure 7c, where the interference peaks are limited up to $550\,\text{cm}^{-1}$. The explanation for this specific behavior stems from the larger electric length of the sample, which leads to increased losses, while such thickness approaches the limits of IRIM. Nonetheless, the estimation is still deemed exceptional, since the transmission line model evaluates a thickness of $98.8 \pm 1.21\,\mu\text{m}$ through the proposed iterative procedure.

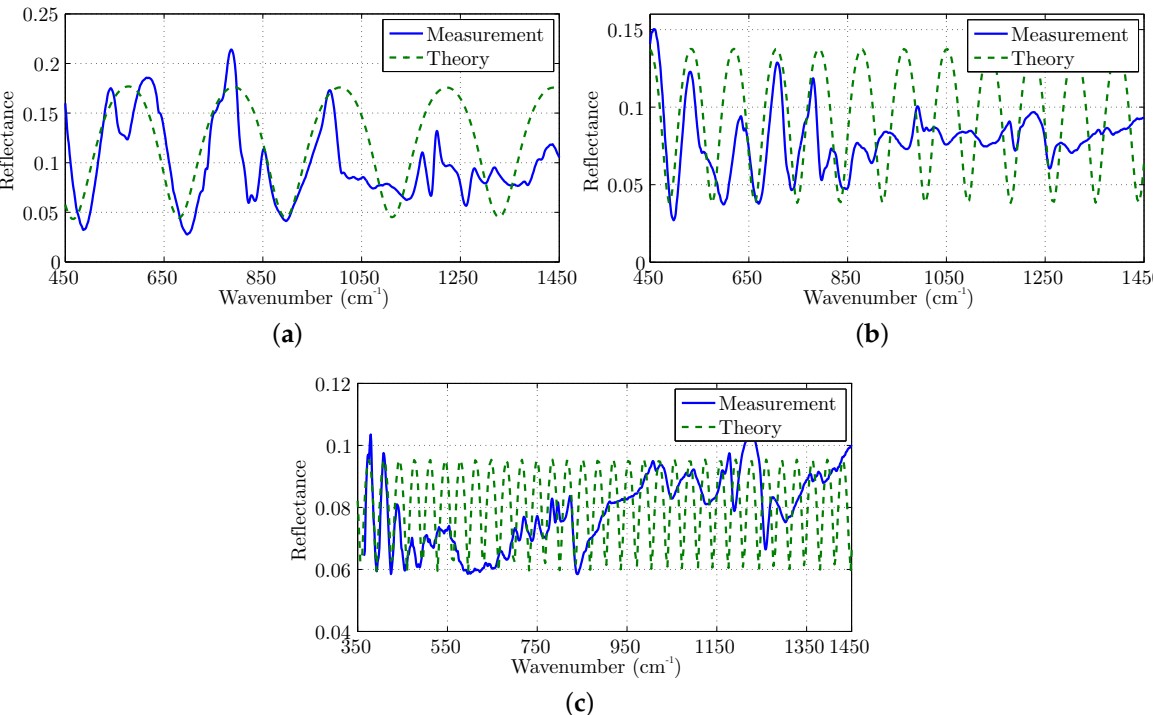

**Figure 7.** Comparison of the proposed transmission line model (theory) to the measured Certified Reference Materials (measurement) with a thickness of (**a**) 15.55 μm, (**b**) 39.71 μm, and (**c**) 98.91 μm.

Furthermore, some real-life coated metallic samples from the "Ormylia" Foundation laboratory were investigated. The electric properties of the applied varnishes are $\varepsilon = 1.96\varepsilon_0$, $\mu = \mu_0$, and $\sigma = 40\,\text{S/m}$, while their thicknesses are at three distinct levels. The metallic substrate is a highly conductive silver alloy, and the reflectance spectrum of an uncoated region is demonstrated in Figure 8. Similarly to the ferrous, the theoretical conductivity of the silver alloy was based on measurement, and its mean value was estimated to be $1.1 \times 10^6\,\text{S/m}$ with $2.7 \times 10^5\,\text{S/m}$ standard deviation, which is close to the expected values in the literature [18]. Note that the conductivity was maintained (at high levels) at any frequency; thus, the near-infrared regime is preferred due to its redundancy-free information nature.

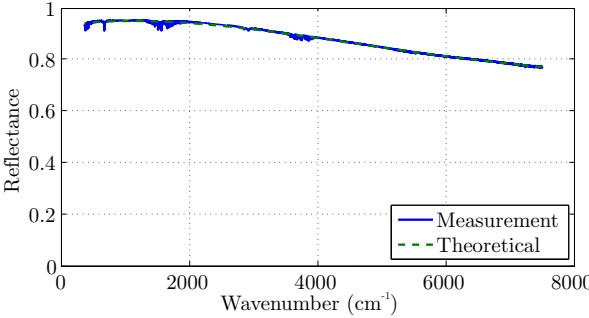

**Figure 8.** The measured versus the theoretical infrared spectrum of an uncoated region of the metallic sample.

The reflectance of the coated regions is depicted in Figure 9 and one can observe that the proposed iterative method, based on the theoretical transmission line approach, approximated the measurements in any case. Specifically, the evaluated thickness of the sample in Figure 9a was $17.5 \pm 0.17\,\mu\text{m}$ as the local extrema alternated quickly. Moreover, in Figure 9b the varnish thickness was calculated to be $6.8 \pm 0.11\,\mu\text{m}$, and the computed reflectance (via the proposed scheme) converged impressively to the measurement. The matching was even better for the sample in Figure 9c, where the thickness was estimated to be $2.41 \pm 0.03\,\mu\text{m}$. The latter indicates that the thinner the layer under investigation,

the more accurate the measurement, and consequently the thickness extraction algorithm. In any case, though, the reliability of the method is remarkable, due to the inspection of the entire spectrum (instead of observing only the extrema, as the conventional IRIM algorithm). Moreover, the thickness can be evaluated reliably in a range from 100 μm to 1 μm for low refractive indices ($n \approx 1.1$), while the range for larger indices (e.g., $n \approx 3$) is from 20 μm to 500 nm. The aforementioned indices correspond to dielectric constants $\varepsilon = 1.2$ and $\varepsilon = 9$, respectively, which are typical values for realistic epitaxial films.

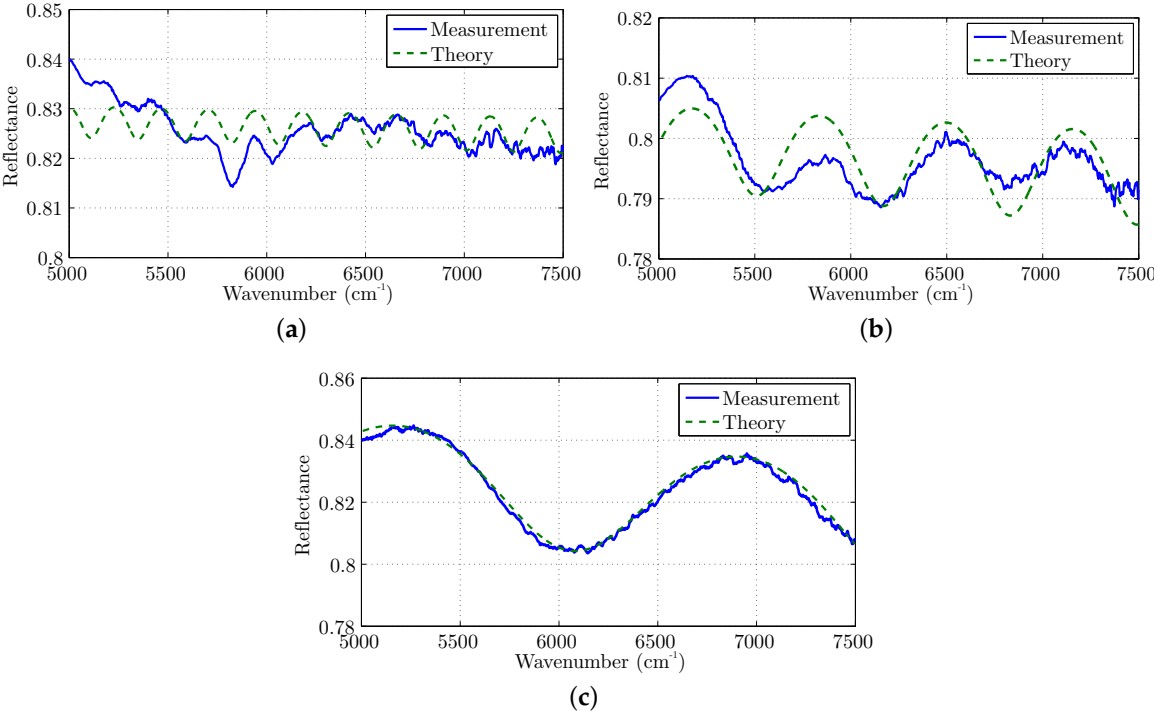

**Figure 9.** Application of the transmission line model to coated metallic samples, to extract thickness (**a**) 17.5 μm, (**b**) 6.8 μm and (**c**) 2.41 μm.

As a future work, additional parameters can be optimized by the genetic algorithm (namely, the electric properties of the varnish). Although they are considered known, small divergences appear in real-life measurements, which confuse the method and delay the convergence. Consequently, electric properties can be introduced into the genetic algorithm by use of constrained bounds.

## 3. Conclusions

A technique for the evaluation of thin-film thickness was introduced in this paper. Extending the primary IRIM concept, the featured scheme takes into account—unlike existing approaches—the multiple reflections of an incident electromagnetic wave, which creates interference on a preselected sensor. Then, the formulation examines the local extrema of the received waveform, in order to estimate the required thickness. For its validation, the results of the proposed method were compared to Certified Reference Materials, exhibiting promising accuracy, consistency, and reliability. Finally, the varnish thickness of coated metallic samples was calculated, utilizing the proposed method.

**Author Contributions:** C.M. and G.A. performed the measurements and the analysis of the spectra. S.A. implemented the theoretical analysis, applied the algorithm at the measured spectra, and wrote the paper. T.Z. and G.K. were the supervisors of the theoretical and experimental procedures, respectively. Finally, N.K. supervised the theoretical analysis and performed the final revision of the paper.

**Acknowledgments:** This work is part of Scan4Reco project, which has received funding from the European Union Horizon 2020 Framework Programme for Research and Innovation under grant agreement no 665091.

**Conflicts of Interest:** The authors declare no conflict of interest. The founding sponsors had no role in the design of the study; in the collection, analyses, or interpretation of data; in the writing of the manuscript; and in the decision to publish the results.

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
