# Peer review of "Development of a Transmission Line Model for the Thickness Prediction of Thin Films via the Infrared Interference Method†"

_technologies, doi:10.3390/technologies6040122_

Round 1
Reviewer 1 Report
This manuscript introduces a technique for the evaluation of thin-film thickness. It is well written and developed.
On page 5, line 117 authors have stated the following: “In particular, a genetic algorithm is utilized in the transmission line model to extract the thickness that minimizes the comparison error.”
Which genetic algorithm is used? Could you explain in more detail the operation of this algorithm? Please provide control parameters (settings) of the optimization algorithm (i.e. number of parameters, population size, minimum and maximum values of optimization variables, etc.)
Author Response
We would like to thank the reviewer for the positive recommendation. Regarding the comment, the genetic algorithm toolbox of MATLAB is used for the optimization and the information of its setup are now included in the paper (lines 119-124)
Reviewer 2 Report
The article has a relevant topic. It's clear and objective.
As suggestions:
Start with the topic introduction
Add suggestions of future work
Author Response
We would like to thank the reviewer for the positive recommendation and the valuable comments that are taken into account to improve the paper quality. Specifically, the introduction is now starting via a brief reference to the topic(lines 13-15), while future suggestions are now added (lines 167-170)
Reviewer 3 Report
Please see the attached file.

Author Response
We would like to thank the reviewer for the valuable comments that enhanced the quality of our work and the generally positive recommendation. All the comments have been seriously considered and successfully performed. In particular, our answers are as follows:
1) The experiments have been repeated as now stated in lines 127-130.The standard deviations for all our measurements are now included
2) The limits of the infrared interference method, relatively to the samples, are now included in lines 169-172.
3) The parameter φ corresponds to the phase shift that is now referenced in line 53
4) The experimental details are now included in lines 127-130. There was not any smoothing algorithm applied, raw data used instead
5) The conductivity has been extracted to fit the measurement in order to facilitate the thickness extraction. Moreover, this value is near the expected through the literature that is now properly cited in line 157.
6) All the related words are now capitalized